# Communicating Personal Melanoma Polygenic Risk Information: Participants’ Experiences of Genetic Counseling in a Community-Based Study

**DOI:** 10.3390/jpm12101581

**Published:** 2022-09-26

**Authors:** Amelia K. Smit, David Espinoza, Georgina L. Fenton, Judy Kirk, Jessica S. Innes, Michael McGovern, Sharne Limb, Erin Turbitt, Anne E. Cust

**Affiliations:** 1The Daffodil Centre, The University of Sydney, A Joint Venture with Cancer Council NSW, Sydney 2000, Australia; 2Melanoma Institute Australia, The University of Sydney, Sydney 2000, Australia; 3Sydney School of Public Health, Faculty of Medicine and Health, The University of Sydney, Sydney 2000, Australia; 4NHMRC Clinical Trials Centre, Faculty of Medicine and Health, The University of Sydney, Sydney 2000, Australia; 5Westmead Clinical School and Westmead Institute for Medical Research, Sydney Medical School, Faculty of Medicine and Health, The University of Sydney, Sydney 2000, Australia; 6Discipline of Genetic Counselling, Graduate School of Health, University of Technology Sydney, Sydney 2000, Australia; 7Maastricht University, 6221 BS Maastricht, The Netherlands; 8Parkville Familial Cancer Centre, Peter MacCallum Cancer Centre and Royal Melbourne Hospitals, Melbourne 3000, Australia; 9Sir Peter MacCallum Department of Oncology, University of Melbourne, Melbourne 3000, Australia

**Keywords:** genetic counseling, polygenic risk score, precision medicine, risk communication, melanoma

## Abstract

Personalized polygenic risk information may be used to guide risk-based melanoma prevention and early detection at a population scale, but research on communicating this information is limited. This mixed-methods study aimed to assess the acceptability of a genetic counselor (GC) phone call in communicating polygenic risk information in the Melanoma Genomics Managing Your Risk randomized controlled trial. Participants (*n* = 509) received personalized melanoma polygenic risk information, an educational booklet on melanoma prevention, and a GC phone call, which was audio-recorded. Participants completed the Genetic Counseling Satisfaction Survey 1-month after receiving their risk information (*n* = 346). A subgroup took part in a qualitative interview post-study completion (*n* = 20). Survey data were analyzed descriptively using SPSS, and thematic analysis of the qualitative data was conducted using NVivo 12.0 software. The survey showed a high level of acceptability for the GC phone call (mean satisfaction score overall: 4.3 out of 5, standard deviation (SD): 0.6) with differences according to gender (mean score for women: 4.4, SD: 0.6 vs. men: 4.2, SD: 0.7; *p* = 0.005), health literacy (lower literacy: 4.1, SD: 0.8; average: 4.3, SD: 0.6; higher: 4.4, SD: 0.6: *p* = 0.02) and polygenic risk group (low risk: 4.5, SD: 0.5, SD: average: 4.3, SD: 0.7, high: 4.3, SD: 0.7; *p* = 0.03). During the GC phone calls, the discussion predominately related to the impact of past sun exposure on personal melanoma risk. Together our findings point to the importance of further exploring educational and support needs and preferences for communicating personalized melanoma risk among population subgroups, including diverse literacy levels.

## 1. Introduction

Melanoma, a deadly form of skin cancer, presents a major challenge to cancer control and public health globally, particularly in fair-skinned populations of European ancestry [1]. There is a need to improve melanoma prevention and early detection, particularly through the better targeting of high-risk groups in the population [2,3]. Precision prevention strategies, including the use of multiple common genomic variants (combined as a polygenic risk score) to improve individual risk assessment and inform risk-specific prevention and early detection, are fast becoming realistic approaches to addressing this need [4].

How to communicate this type of personalized health information to individuals in the community is an important consideration for the successful implementation of polygenic risk scores in precision prevention. Another important consideration is understanding the role health professionals should play in results disclosure and follow-up. Traditional models for genetic testing typically involve time-intensive, in-person pre- and post-test counseling with a genetics-trained health professional (e.g., a genetic counselor) in a tertiary care setting. In this traditional context, genetic counseling consultations aim to support high-risk families with informed decision-making and managing psychosocial and familial impacts that may arise from testing for high penetrance single genes (e.g., CDKN2A) [5]. By contrast, polygenic risk information is not predictive of family members’ risk, and it usually confers lower absolute risk compared to rarer high penetrance single genes. and applies to the general population, not just high-risk families [6,7,8]. Receiving polygenic cancer risk information has been demonstrated to not cause adverse psychological outcomes [9]. Given these differences, communicating this type of information to the community may be better suited to alternative modes that are less resource-intensive and more accessible than traditional genetic counseling [10,11].

In the context of delivering polygenic risk information, traditional genetic counseling consultations may be best reserved for patients with more complex and uncertain situations where detailed information, education, and psychological support can be most beneficial [12]. Alternative modes include telephone genetic counseling, return of results by mail [13], and direct deposit of results into a patient’s Electronic Medical Record. Other deviations from traditional genetic counseling could include altering the focus of result delivery sessions to concentrate on education and motivating behavior change (i.e., prevention-focused counseling). There are gaps in evidence on how prevention-focused counseling should be incorporated into precision prevention strategies, including the acceptability of this approach to the broader population, and within specific population subgroups, which is an important consideration for successful implementation [14].

In this study, we examined the acceptability of the genetic counselor phone call in the Managing Your Risk Study (a randomized controlled trial that returned personalized melanoma polygenic risk information to the Australian community). We measured participant satisfaction and tested whether this differed according to individual characteristics such as age, gender, personal risk of melanoma, and health literacy/numeracy. We also examined key topics arising during the phone call and fidelity with the communication manual. These findings will inform future research on the development of educational and support resources for communicating the polygenic risk of information to the community.

## 2. Materials and Methods

### 2.1. Study Design and Participants

We used a concurrent, mixed-methods design [15]. For this study, we have focused on data pertaining to the genetic counseling phone call that was part of the intervention used in the Australian Melanoma Genomics ‘Managing Your Risk Study’ randomized controlled trial. The design and main findings from the Managing Your Risk Study have been previously published [16,17].

Briefly, participants (*n* = 1025) from all States and Territories in Australia were recruited via Medicare (Australia’s publicly funded universal healthcare system). Eligible participants had no personal history of melanoma, some European ancestry, and were aged between 18–69 years. After baseline data collection (online questionnaire and wearing an objective measure of sun exposure, a dosimeter, for 10 days), participants were randomized 1:1 to the intervention or control arm. The intervention group (*n* = 513) was mailed a saliva collection kit for DNA collection and genotyping (successfully completed by *n* = 509) and mailed their personal polygenic risk of melanoma information and educational information on melanoma prevention and early detection in hardcopy booklets; participants in the control arm were provided with the educational information booklet only (see Figure 1 below). The polygenic risk score was based on 40 variants (single nucleotide polymorphisms; SNPs) in 20 genes/gene regions with established associations with melanoma risk [18] and was presented as a risk category (lower than average, average, higher than average) and an absolute remaining lifetime risk in a pictograph format [17].

Within two weeks of mailing the intervention materials, the study genetic counselor contacted all intervention participants (*n* = 509) via telephone and these conversations were guided by a communication manual (adapted from our previous research [19]). The manual had a script and checklist for the genetic counselor to follow, including confirming whether the participant had received their booklets and whether they could recall their risk category, asking how they felt about their personal risk information, and answering any participant questions (Figure 1).

If the participant had not yet received their booklets, the genetic counselor offered to provide their risk information during the phone call and re-post their hardcopy booklets. The telephone calls were audio recorded and as part of the checklist, the genetic counselor documented the topics raised by participants.

We used quantitative methods to assess participant satisfaction with the genetic counseling phone call and to check the fidelity between the phone call and the communication manual, and qualitative methods to identify the key topics arising during the phone calls and interviews. We used data triangulation by reporting and interpreting the findings from qualitative and quantitative analysis together [20]; this method is appropriate where understanding the context of participant experiences is part of the research aims [21]. We followed standard reporting guidelines for the analysis and reporting of mixed research [22].

Ethics approval was obtained from The University of Sydney and written informed consent was obtained from all participants.

### 2.2. Data Collection

Participant satisfaction with the genetic counseling phone call was measured using a modified version of the Genetic Counseling Satisfaction Survey [23] at 1-month post-intervention delivery via an online questionnaire. All intervention participants who received their personal risk information (*n* = 509) were asked “have you spoken to the genetic counselor since receiving your risk information?” and if they responded, ‘Yes’ or ‘I’m not sure’ they were prompted to complete the Genetic Counseling Satisfaction Survey for which they were asked to rate six-items on a five-point Likert scale (strongly disagree = 1 to strongly agree = 5). The first two items in the original scale were modified to be relevant to our study design and context: “My genetic counselor seemed to understand the stresses I was facing” was changed to “I felt I could talk about my reaction to my risk information with the genetic counselor”; and, “My genetic counselor helped me to identify what I needed to know to make decisions about what would happen to me” was modified to “The genetic counselor helped me to understand my risk information and make decisions about my health care”.

The key topics arising during the phone call and all participant questions were collected by the genetic counselor using the communication checklist and calls were audio-recorded. Phone call fidelity checks were performed by two members of the research team (AKS, ET), which involved listening to a sample of genetic counselor phone call recordings and cross-checking the content delivered by the genetic counselor with the manual, including the completion of the checklist and documentation of discussion themes [24]. The phone call recordings for the fidelity checks were purposively sampled to ensure that a range of genomic risk categories was included (6 = low, 5 = average, 6 = high) and the sample was balanced for gender.

Participant satisfaction with the genetic counseling phone call was also explored qualitatively during semi-structured interviews with a sub-group of intervention arm participants after completion of the Managing Your Risk Study (*n* = 20). During the interviews, participants were asked about their experience in the larger study including: “How did you feel about then receiving a phone call from a genetic counselor to discuss your results?”. The interviews were audio recorded and professionally transcribed. The interview data presented here are specific to participant responses to this question about the genetic counselor phone call.

On the Participant Information Statement, participants were informed that they had the option of speaking with the genetic counselor about their decision to participate in the study and any calls were noted in the research study database.

### 2.3. Data Analysis

Questionnaire responses to the Genetic Counseling Satisfaction Survey were descriptively analyzed to generate mean and standard deviation values of the ratings for each item and the overall score. Bivariate and multivariate analyses were performed to investigate if mean satisfaction ratings by each statement in the survey differed across participant characteristics including age group (18–44 years vs. 45–69 years), gender (male vs. female), polygenic risk of melanoma (lower than average, average, higher than average) health literacy (measured by: *How confident are you filling out medical forms by yourself?* Response options grouped as: *not at all/a little bit/somewhat, quite a bit, extremely* [25]), health numeracy (measured by: *In general, how easy or hard do you find it to understand information that has a lot of numbers and statistics?* Response options grouped as: *very easy, easy, hard or very hard* [25]), and children (yes/no). Significant differences in mean scores were determined by performing an independent sample t-test or ANOVA. Statistical analyses were performed using IBM SPSS Statistics Software (Version 27); statistical significance was assessed at the 0.05 level.

Qualitative data from the interviews and the genetic counselor’s phone call notes were analyzed using coding reliability thematic analysis [26,27]. The data were coded based on a predetermined set of codes, reviewed, and further refined into broader categories that would guide the development of topic summaries, using NVivo 12.0 software. We aimed to develop topic summaries that were relevant to receiving personal genomic risk information and the provision of genetic counseling, including areas of concern or confusion.

We integrated quantitative and qualitative data relating to participant satisfaction by building on the questionnaire findings and further exploring participant experiences through the semi-structured interviews.

## 3. Results

### 3.1. Participants

Of the 509 participants who provided a saliva sample, the genetic counselor spoke with 448 (88%) via telephone and communicated with 12 (2%) via email only. The remaining participants (*n* = 49, 10%) did not respond to three phone call attempts or emails from the genetic counselor. The average length of the phone calls was 7 min (range: 2–25 min). Twenty participants took part in a qualitative interview post-completion of the parent study. Participant characteristics are summarized in Table 1.

### 3.2. Satisfaction with the Genetic Counseling Phone Call

In the follow-up questionnaire (sent to participants one-month post-intervention delivery), 336 (66%) reported that they spoke with the genetic counselor, 10 (2%) were unsure, 150 (29%) reported that they did not speak with the genetic counselor and 13 (3%) did not complete the questionnaire. Among those participants who reported that they did not speak with the genetic counselor, 84 identified as male (aged 18–44 years = 29, 45–69 years = 55) and 66 as female (18–44 years = 34, 45–69 years = 32). All participants who responded yes or unsure to this question completed the Genetic Counseling Satisfaction Scale one month after receiving their booklets (Table 2).

The overall mean score was 4.3 standard deviation (SD): 0.63. The individual item with the highest mean score overall was ‘I felt I could talk about my reaction to my risk information with the genetic counselor’ (mean, M = 4.7, SD = 0.66) followed by ‘the length of the phone call was appropriate’ (M = 4.6, SD = 0.66) and ‘the genetic counselor helped me to understand my risk information and make decisions about my health care’ (M = 4.5, SD = 0.73). The item with the lowest mean score overall was ‘I felt better about my health after talking to the genetic counselor’ (M = 4.0, SD = 0.89).

The Genetic Counseling Satisfaction Scale total mean scores differed according to some participant characteristics (Table 2). Women tended to have higher scores (indicating higher satisfaction) compared to men (*p* = 0.0063), and participants with higher health literacy had higher scores compared to those with lower health literacy (*p* = 0.024). According to the polygenic risk category, participants with low risk had the highest overall mean score followed by average and high risk (*p* = 0.03). The mean score for the individual item ‘the genetic counselor was truly concerned about my well-being’ was higher for older participants compared to younger participants (*p* = 0.038), and for participants with children compared to those without children (*p* = 0.005). Total mean and individual item scores did not differ according to family history of melanoma, personal history of other skin cancer, or health numeracy. 

Satisfaction was explored in the qualitative interviews to further contextualize the questionnaire findings. Participants who described positive experiences with the genetic counseling call felt that it had provided them with an opportunity to ask questions and clarify their personal risk information:

“I think the one-on-one contact and to be able to listen to a professional talk about the topic was very good and not being [a] medical, scientific person [myself], it’s good to hear that and [it] gave you a chance to ask a couple of questions along the way.” (Male, 59 years, high risk)

Other participants described more neutral views towards the genetic counseling call:

“Yeah, it [the phone call] was fairly short and didn’t really take much time out of my day” (Male, 21 years, high risk)

Another key topic from the interviews was that participants felt the genetic counseling call would be beneficial for those receiving high-risk results but not necessarily lower-risk results, and this view was particularly evident among participants at low or average risk:

“I don’t remember what we even said, to be honest. I think they asked if I had any questions or if I wanted any advice (…) because my results were that my risk is a little bit lower than average, it wasn’t stressful for me, but I’m sure if it was a little bit higher than normal, then I probably would have had some questions.” (Female, 25 years, low risk)

“I didn’t feel I needed the phone call, but I was happy to have the phone call, because the book was pretty self-explanatory, and I went through it all. So, I didn’t feel that added any value. I think if my results had been higher risk, then the phone call would have been useful, but my risk came out lower, so I was pretty relaxed.” (Female, 46 years, low risk)

### 3.3. Discussion Topics and Participant Questions during the Genetic Counseling Phone Call

The genetic counselor documented all participant questions and discussion areas, which were grouped by topic (Table 3). The most prevalent discussion topics were related to personal risk information, including ‘I have had a lot of sun exposure when I was younger. How does this influence my personal risk?’ (20%) and ‘How was my risk information calculated?’ (15%). Participants frequently referred to their sun exposure, both past and current sun habits, in understanding their personal risk information, and some thought that their past sun habits were included in their risk estimate.

As part of the communication manual, the genetic counselor clarified that the risk information did not include past sun exposure or sunburns and therefore their actual risk could be higher than the provided estimate, which was only based on genomic variants, age, gender and place of residence.

Other common discussion topics included ‘How do the 20 genes that are examined relate to melanoma development?’ (10%) in which participants were interested in the amount of risk conferred by their genes, and relatedly, some participants asked about the range of risk in the population including what comprises a high-risk estimate.

### 3.4. Fidelity between the Genetic Counseling Phone Call and Communication Manual

The majority of the genetic counseling phone calls were audio recorded (*n* = 352, 79%). Fidelity checks (completed for 17 phone call recordings, 5%) showed that 91% of the communication manual domains were addressed by the genetic counselor during the phone calls (Appendix A). The checklists maintained by the genetic counselor during the calls correctly corresponded to each participant’s telephone session.

### 3.5. Uptake of Genetic Counseling Access Outside of Scheduled Calls

During recruitment, participants were provided with the option of speaking with the genetic counselor, which was taken up by three participants who asked questions related to life insurance implications of receiving the risk information. Outside of the scheduled calls during the study, no participants requested to speak with the genetic counselor–although this was offered to participants on the Participant Information Statement at the start of the study and again during the genetic counselor phone call.

## 4. Discussion

Our findings show that the Australian community had high satisfaction with a genetic counseling phone call after receipt of information on personal polygenic risk of melanoma alongside educational information on melanoma prevention and early detection. Patient satisfaction with healthcare has been described as a comparison between patient expectations for a service or treatment and what they receive [23]. Satisfaction also relates to acceptability (i.e., the perception that a given practice is agreeable to key stakeholders), which is an important consideration for implementation [14]. The phone call provided an important opportunity for participants who had questions about their personal risk to discuss these with the genetic counselor, although many felt that the genetic counseling phone call would be most beneficial for those who receive a high polygenic risk estimate. We observed variations in levels of satisfaction and in discussion topics that highlight some key considerations relevant to scaling up this approach to providing personal risk information to the community, which are discussed below.

Satisfaction has been widely used to evaluate the quality of genetic counseling in specialty clinic settings for individuals at high risk of a monogenic condition [28], and telephone or telehealth genetic counseling has been shown to be highly satisfactory for patients in this setting [29,30]. Comparatively, the role of genetic counseling in the context of providing complex genomic risk information to the broader community has been less explored. We found that participant satisfaction with the genetic counseling process in the Managing Your Risk Study, while overall high, was lower for men compared to women, those with lower health literacy levels compared to higher, and varied according to polygenic risk results (with high-risk groups experiencing lower satisfaction). Some participants also reported that they did not speak with the genetic counselor in the follow-up questionnaire (*n* = 150, 29%). Over half of these participants who reported not speaking with the genetic counselor identified as male (*n* = 84) and were mainly in the older age group (*n* = 55). Together these findings support the need to further investigate potential educational and support needs for men, groups with lower health literacy, and recipients of high-risk results to ensure acceptability for these population subgroups. Kaphingst *et al.* has demonstrated that groups with lower health literacy skills tend to have significantly lower genetic knowledge compared to higher literacy levels [31] and experience higher levels of confusion in response to receiving personal melanoma genomic risk information [32]. Strategies to lower the health literacy demands of personal genomic risk results, such as additional interpersonal sources of information with a trained health educator for example, have been suggested [32].

Limited genetic counselor workforce capacity and costs of service delivery are driving the development of alternative delivery models for genetic and genomic information that are more equitable and accessible [33]. These alternative models include the use of mailed results, online portals and chatbots [34] that are less resource-intensive compared to traditional approaches. A randomized controlled trial in an Australian general practice demonstrated the feasibility and acceptability of providing genomic risk of colorectal cancer and corresponding screening advice [7]. The offer for testing and delivery of results was provided by a genetic counselor and followed a concise standardized script that was designed to be adopted by a general practitioner in the future.

Non-genetics health professionals could be trained to offer testing and deliver results to assist in managing the volume and scale required for population-level testing. In two randomized controlled trials in primary care settings in the US, intervention arm participants were posted hardcopy booklets presenting their personal melanoma genomic risk information and educational information on prevention and early detection, followed by a phone call by a trained researcher or health educator (as opposed to a genetic counselor) [35,36]. Participant satisfaction with these study processes was high overall.

Our finding that conversations with the study genetic counselor focused on past sun exposure, preventive behaviors and melanoma risk factors supports the provision of this type of information via other (non-genetics specialist) health professionals. Participants in our study noted, speaking with a genetic counselor may be preferred for providing high risk results. We found that people very closely equate their past sun exposure and current sun habits with their personal risk of melanoma, which is not surprising given that sun exposure is the strongest risk factor for melanoma and sun protection is widely promoted in Australia through well-established public health campaigns. Much of the conversations with the study genetic counselor were spent explaining the impact of sun exposure on their personal risk, and that it may be higher than the provided estimate. That individuals may misattribute sun exposure to personal genetic risk highlights a potential educational need for delivering melanoma genomic risk information in future programs. Our finding that the discussion with the genetic counselor focused on risk factors and preventive behaviors may be translatable to communicating polygenic risk information for other complex, preventable conditions (such as other cancers or coronary artery disease) for which modifiable risk factors that are not captured in a polygenic risk score may impact individual risk, such as lifestyle and diet.

Supporting preventive behavior change measures at the population level aligns with a broader shift in role diversification for genetic counselors [37]. This role diversification represents a fundamental change in practice from a specialized clinical model to a public health (prevention) focus. Alongside this shift from a clinical to public health focus is a need to adapt ethical frameworks underlying genetic counseling practice. For example, Schupmann et al. [38] advocate for a move away from non-directiveness in genetic counseling towards a framework that highlights beneficence, non-maleficence, and justice. Similarly, Newson argues that there are deficiencies in applying individualized medical ethics concepts such as autonomy to a population-level program and that ethical frameworks that incorporate collectivism and solidarity may be more appropriate [39].

Preferences for the type of health professional offering and providing polygenic risk information, and the nature of the educational and support resources, may also vary across diverse populations, cultural and linguistic groups, which was not explored in our study [40]. For example, a qualitative interview study with diverse Spanish- and English-speaking patients on the clinical use of polygenic risk scores for hypocholesterolemia showed that the majority of participants favored in-person disclosure, followed by electronic and telehealth delivery [41]. Few participants preferred receiving polygenic risk information by mail or by telephone [41]. Future studies could further explore satisfaction and perceived acceptability of communication modes and support resources that are tailored to the needs and preferences of diverse populations to ensure successful and equitable implementation of polygenic risk information into clinical practice [42].

This study explored the experiences of receiving genetic counseling in a setting that is yet to be widely practiced in this health profession. The large sample size and mixed methods approach, which exploits the advantages of both qualitative and quantitative approaches are further strengths. Although the dataset used in this study was considerable in size and participants were randomly selected, these participants are likely to be more interested in genomic testing compared to the broader population. Most participants had a high level of health literacy, numeracy and education, which also limits our ability to compare our results across these groups. As per the eligibility requirements for the larger study, all participants had European ancestry and there is a need to extend the inclusion of more diverse populations in future research.

## 5. Conclusions

Our findings suggest a model of telephone genetic counseling in a polygenic testing context was well-received by participants drawn from the Australian community. Differences in satisfaction across literacy levels, gender and polygenic risk groups suggest there are opportunities to further explore the needs and concerns of certain groups in order to provide improved support in a setting that is expected to become increasingly common in the future of genetic counseling practice.

## Figures and Tables

**Figure 1 jpm-12-01581-f001:**
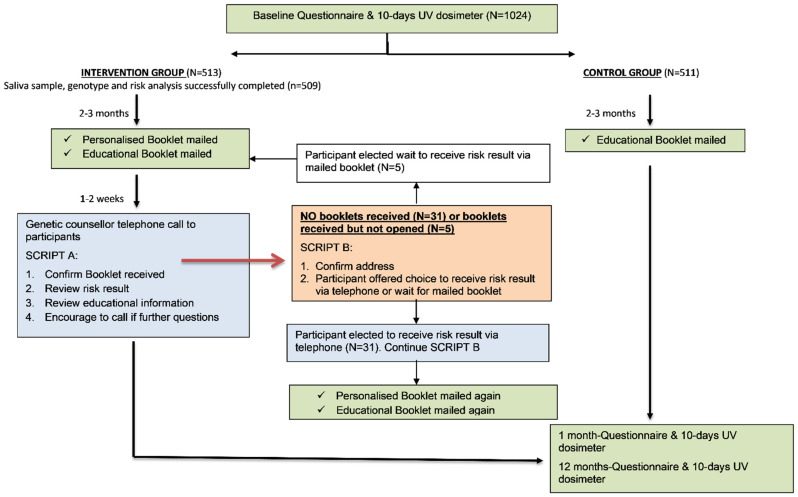
Participant flow for the provision of the intervention and control materials in the Melanoma Genomics Managing Your Risk Study.

**Table 1 jpm-12-01581-t001:** Participant characteristics.

Characteristics	Intervention Arm Participants (N = 513)	Interviewed Participants from the Intervention Arm (N = 20)
	N (%)	N (%)
Females N, (%)	261 (50.9%)	11 (55.0%)
Age group N, (%)		
18–44 years	241 (47.0%)	9 (45.0%)
45–69 years	272 (50.3%)	11 (55.0%)
Polygenic risk category		
Lower than average	107 (20.9%)	10 (50.0%)
Average	267 (52.0%)	3 (15.0%)
Higher than average	135 (26.3%)	7 (35.0%)
Country of birth		
Australia	409 (79.7%)	17 (85.0%)
New Zealand	20 (3.9%)	1 (5.0%)
United Kingdom	40 (7.8%)	1 (5.0%)
Other	44 (8.6%)	1 (5.0%)
Marital status		
Never married	99 (19.3%)	4 (20.0%)
Widowed	6 (1.2%)	1 (5.0%)
Separated or divorced	39 (7.6%)	1 (5.0%)
Married or in a de facto relationship	369 (71.9%)	14 (70.0%)
Highest level of education		
Primary school (or equivalent)	2 (0.4%)	0
High school (or equivalent)	118 (23.0%)	4 (20.0%)
Certificate/diploma	176 (34.3%)	6 (30.0%)
University degree	217 (42.3%)	10 (50.0%)
Health literacy ^1^		
Extremely confident, quite a bit confident	447 (57.1%)	18 (90.0%)
Not at all, a little bit, or somewhat confident	66 (12.9%)	2 (10.0%)
Health numeracy ^2^		
Very easy, Easy	412 (80.3%)	18 (90.0%)
Very hard, Hard	101 (19.7%)	2 (10.0%)
Previously diagnosed with non-melanoma skin cancer		
No	411 (80.1%)	16 (80.0%)
Yes	83 (16.2%)	3 (15.0%)
I don’t know	63 (12.3%)	1 (5.0%)
First-degree blood relatives with a previous melanoma		
No	353 (68.8%)	16 (80.0%)
Yes	97 (18.9%)	3 (15.0%)
I don’t know	63 (12.3%)	1 (5.0%)

^1^ Health literacy was measured by asking “How confident are you filling out medical forms by yourself?” ^2^ Health numeracy was determined by asking “In general, how easy or hard do you find it to understand medical statistics?”.

**Table 2 jpm-12-01581-t002:** Genetic Counseling Satisfaction Survey (GCSS) mean scores overall and within subgroups.

GCSS Items	Overall	Age Group		Gender		Children		Health Literacy			Polygenic Risk Category		
	(*n* = 346) ^1^	18–44 yrs (*n* = 164)	45–69 yrs (*n* = 182)	Female (*n* = 187)	Male (*n* = 159)	Yes (*n* = 241)	No (*n* = 105)	Extremely (*n* = 217)	Quite a Bit (*n* = 91)	Not/Little/Somewhat (*n* = 38)	Lower (*n* = 78)	Average (*n* = 180)	Higher (*n* = 88)
	*M (SD)*	*M (SD)*	*M (SD)*	*M (SD)*	*M (SD)*	*M (SD)*	*M (SD)*	*M (SD)*	*M (SD)*	*M (SD)*	*M (SD)*	*M (SD)*	*M (SD)*
I felt I could talk about my reaction to my risk information with the genetic counselor	4.70 (0.66)	4.65 (0.72)	4.74 (0.60)	4.74 (0.62)	4.65 (0.70)	4.71 (0.61)	4.68 (0.78)	**4.77 (0.58)**	**4.67 (0.62)**	**4.34 (1.02) ^2^**	4.81 (0.40)	4.67 (0.63)	4.66 (0.87)
The genetic counselor helped me to understand my risk information and make decisions about my health care	4.47 (0.73)	4.46 (0.72)	4.49 (0.74)	4.52 (0.70)	4.42 (0.77)	4.50 (0.71)	4.41 (0.78)	4.53 (0.72)	4.43 (0.67)	4.24 (0.88)	**4.67 (0.53)**	**4.43 (0.72)**	**4.39 (0.88)** ** ^3^ **
I felt better about my health after talking to the genetic counselor	4.02 (0.89)	4.01 (0.96)	4.03 (0.82)	**4.11 (0.81)**	**3.91 (0.96) ^4^**	4.03 (0.92)	4.00 (0.83)	4.00 (0.92)	4.14 (0.80)	3.84 (0.92)	**4.29 (0.69)**	**3.99 (0.90)**	**3.83 (0.97) ^5^**
The length of the phone call was appropriate	4.58 (0.66)	4.59 (0.67)	4.57 (0.65)	**4.67 (0.57)**	**4.48 (0.75) ^6^**	4.59 (0.63)	4.55 (0.72)	4.64 (0.63)	4.52 (0.67)	4.42 (0.76)	4.72 (0.48)	4.53 (0.67)	4.56 (0.76)
The genetic counselor was truly concerned about my well being	4.12 (0.95)	**4.01 (0.98)**	**4.23 (0.92) ^7^**	**4.25 (0.87)**	**3.98 (1.0) ^8^**	**4.22 (0.90)**	**3.90 (1.04) ^9^**	**4.20 (0.91)**	**4.12 (0.98)**	**3.71 (1.04) ^10^**	4.23 (0.91)	4.04 (1.01)	4.19 (0.86)
Talking to the genetic counselor was valuable to me	4.11 (0.92)	4.01 (0.99)	4.20 (0.84)	**4.24 (0.85)**	**3.96 (0.98) ^11^**	4.14 (0.90)	4.03 (0.96)	4.12 (0.95)	4.15 (0.84)	3.89 (0.92)	4.28 (0.79)	4.02 (0.97)	4.14 (0.90)
Total mean score	4.33 (0.63)	4.29 (0.67)	4.38 (0.59)	**4.42 (0.55)**	**4.23 (0.70) ^12^**	4.37 (0.63)	4.26 (0.63)	**4.38 (0.61)**	**4.34 (0.62)**	**4.08 (0.76) ^13^**	**4.50 (0.48)**	**4.28 (0.65)**	**4.29 (0.70) ^14^**

^1^ Participants were asked “have you spoken to the genetic counselor since receiving your risk information?” and if they responded ‘Yes’ (*n* = 336) or ‘I’m not sure’ (*n* = 10) they were prompted to complete the GCSS. Bold text indicates a significant difference between in mean GCSS scores across groups. ^2^ *p*-values obtained from independent samples t-test or ANOVA. *p* = 0.001; ^3^ *p* = 0.026; ^4^ *p* = 0.039; ^5^ *p* = 0.0028; ^6^ *p* = 0.0087; ^7^ *p* = 0.038; ^8^ *p* = 0.0109; ^9^ *p* = 0.0054; ^10^ *p* = 0.0142; ^11^ *p* = 0.0052; ^12^ *p* = 0.0063; ^13^ *p* = 0.024; ^14^ *p* = 0.0299.

**Table 3 jpm-12-01581-t003:** Participant questions grouped by topics, for those who spoke with the genetic counselor via phone.

Discussion Topic	General Questions	Number of Related Questions (*n* = 448)	Example Participant Quotes
Risk result information	I have had a lot of sun exposure when I was younger. How does this influence my personal risk?	89 (20%)	“I grew up in the times when you just baked in the sun when you were a kid. I don’t now, but I have damage, there’s no doubt about that” (Female, 65 years, low risk)
	How was my risk information calculated?	68 (15%)	“How are all the genes, such as the repair genes, included in the risk equation?” (Male, 63 years, average risk)
	How do the 20 genes that are examined relate to melanoma development?	47 (10%)	“I’m at low risk. Does this mean that I have no genetic risk, only an environmental risk?” (Female, 65 years, low risk)
	What kind of risk information does the genetic result provide?	11 (2%)	“I wonder, what number is high risk? I look at some of my friends and think, you must be at 20%” (Male, 43 years, average risk)
	How does my ancestry influence my risk information?	10 (2%)	“Did you categorize the groups based on where the DNA shows where you come from?” (Male, 63 years, average risk)
	Does the testing look at other genes for other conditions?	10 (2%)	“What other information can be derived from my DNA sample?” (Male, 42 years, high risk)
	I have a strong family history of melanoma or other skin cancers. What does this mean for my risk?	9 (2%)	“I’m surprised by my low risk because my sister has had loads of melanoma cut out in her late 20 s, but they may have been lower grade ones” (Female, 44 years, average risk)
	Could my genetic risk result be related to other cancers I have had or are in my family?	5 (1%)	“Do these genes relate to other cancers or only melanoma?” (Female, 65 years, low risk)
Healthcare/prevention & early detection of melanoma	Can my GP do a skin check or do I need to go to a specialist skin cancer clinic or dermatologist?	41 (9%)	“The quick option is going to the GP. But I might as well go to a specialist, although sometimes you have to wait for a month” (Male, 43 years, average risk)
	How often do I need a professional skin check?	35 (8%)	“Even though I am low risk and I’m not freckly, would you recommend having a health professional check my skin?” (Female, 47 years, low risk)
	Can my results be given to my GP or dermatologist?	19 (4%)	“I don’t have a regular GP, I haven’t been to a doctor for a long time but there’s a skin cancer clinic where I go for skin checks and I can show them my risk information.”(Male, 35 years, low risk)
	I thought it was important for me to get some sun to maintain adequate Vitamin D levels?	9 (2%)	“Do you still get Vitamin D when you’ve got sunscreen on?” (Male, 36 years, average risk)
	What can I do to reduce my risk of melanoma?	11 (2%)	“Should we just avoid the sun during 11-3 pm altogether?” (Female, 37 years, average risk)
	How can I reduce my risk of melanoma using sunscreen?	10 (2%)	“What are the best sunscreens? For example, what are the differences between normal sunscreen and those that you can buy in health food shops?” (Female, 37 years, average risk)
	How do I check myself for melanoma?	3 (1%)	“I have never done a skin check before, but I will discuss it with my GP” (Male, 42 years, average risk)
Familial risk/risk for other family members	What will the results mean for my children?	16 (4%)	“How relevant are my results for my children?” (Female, 47 years, low risk)
	Does this affect other family members (siblings or parents)?	14 (3%)	“What are the implications for my family?” (Male, 43 years, average risk)
	Should I inform my children or other family members of this risk information?	4 (1%)	“I will tell my children as I imagine their risk is just over the line like mine, but they haven’t been sunburnt” (Female, 56 years, high risk)
Life insurance	Will this affect my ability to obtain life insurance?	1 (0%)	“Do you happen to know whether average risk is considered a problem with people getting life insurance, or is it more to do with people at high risk?”(Female, 40 years, average risk)
General study information	What was the purpose of this study?	36 (8%)	“Where will the findings from this study lead?” (Male, 69 years, high risk)
	What information does the ultraviolet (UV) dosimeter provide?	37 (8%)	“How does the wristband actually work?” (Female, 55 years, average risk)
	Who is being recruited into the study?	21 (5%)	“I wondered how you selected people to take part in the study?” (Female, 55 years, average risk)
	What will happen to my saliva/DNA sample after the study?	3 (0%)	“What happens to the DNA sequence, is that stored on your servers?” (Male, 42 years, high risk)

## Data Availability

De-identified data from this analysis is available on request by contacting the corresponding author. Ethics approval to share the data will be required.

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
