# Peer review of "Communicating Personal Melanoma Polygenic Risk Information: Participants’ Experiences of Genetic Counseling in a Community-Based Study"

_jpm, 2022, doi:10.3390/jpm12101581_

Round 1
Reviewer 1 Report
This is a finely designed and executed study and the manuscript is well written and detailed enough to present conclusions. The application of this study is narrow, as it deals with one disease entity and very specific population, which does not take away of its importance. I wish to see a few additional sentences on how this can be extended to other populations and other diseases within the group. Also, the addition of some advice and guidance in the discussion for other cultural entities and populations to conduct their own studies. This would increase the value of the study, albeit it is excellent as is.
Author Response
In the Discussion, we have provided additional sentences to address these points raised by Reviewer #1, page 5:
“Our finding that the discussion with the genetic counselor focused on risk factors and preventive behaviors may be translatable to communicating polygenic risk information for other complex, preventable conditions (such as other cancers or coronary artery disease) for which modifiable risk factors that are not captured in a polygenic risk score may impact individual risk, such as lifestyle and diet…
Preferences for the type of health professional offering and providing polygenic risk information, and the nature of the educational and support resources, may also vary across diverse populations, cultural and linguistic groups, which was not explored in our study [40]. For example, a qualitative interview study with diverse Spanish- and English-speaking patients on the clinical use of polygenic risk scores for hypocholesterolemia showed that the majority of participants favored in-person disclosure, followed by electronic and telehealth delivery [41]. Few participants preferred receiving polygenic risk in-formation by mail or by telephone [41]. Future studies could further explore satisfaction and perceived acceptability of communication modes and support resources that are tailored to the needs and preferences of diverse populations to ensure successful and equitable implementation of polygenic risk information into clinical practice [42].”
Reviewer 2 Report
The prevalence and mass character of genetic tests leads to the fact that the medical community must look for new ways to communicate genetic information. This work is not the first in which genetic information about risks is reported by phone. This study has once again shown the acceptability of using various ways of communicating information when using good accompanying materials and competently written scripts. I have no significant comments on the work and its presentation. I suggest that part of the information concerning the size of the surveyed groups that received the call and the questionnaire be duplicated in the text of the materials and methods section, and not only in Figure 1.
I consider this work worthy of publication. The accumulation of such mass research will allow the healthcare system to adapt to the challenges of our time.
Author Response
We have addressed this suggestion by Reviewer #2 in the Materials and Methods section, page 3: “Within two weeks of mailing the intervention materials, the study genetic counselor contacted all intervention participants (n=509) via telephone”
And page 4: “All intervention participants who received their personal risk information (n=509) were asked “have you spoken to the genetic counselor since receiving your risk information?” and if they responded, ‘Yes’ or ‘I’m not sure’ they were prompted to complete the Genetic Counseling Satisfaction Survey for which they were asked to rate six-items on a five-point Likert scale (strongly disagree = 1 to strongly agree = 5).”
Reviewer 3 Report
The authors performed a study regarding the communication of personalized melanoma polygenic risk information and the satisfaction/understanding of the participants. The study design is adequate, their results support their conclusions. The authors acknowledged the limitations of their study and suggested further research directions regarding melanoma risk among Australian populations.
Author Response
No changes have been made in response to this feedback.